# Can Circulating MicroRNAs, Cytokines, and Adipokines Help to Differentiate Psoriatic Arthritis from Erosive Osteoarthritis of the Hand? A Case–Control Study

**DOI:** 10.3390/ijms26104621

**Published:** 2025-05-12

**Authors:** Antonella Fioravanti, Sara Cheleschi, Etienne Cavalier, Jean-Yves Reginster, Majed Alokail, Aurélie Ladang, Sara Tenti, Giorgio Bedogni

**Affiliations:** 1Independent Researcher, 53100 Siena, Italy (Previously Responsible for the Clinic for the Diagnosis and Management of Hand Osteoarthritis, of the Rheumatology Unit, Department of Medicine, Surgery and Neuroscience, Azienda Ospedaliera Universitaria Senese, Policlinico Le Scotte, 53100 Siena, Italy); 2Rheumatology Unit, Department of Medicine, Surgery and Neuroscience, Azienda Ospedaliera Universitaria Senese, Policlinico Le Scotte, 53100 Siena, Italy; saracheleschi@hotmail.com; 3Department of Clinical Chemistry, CHU de Liège, University of Liège, 4000 Liege, Belgium; etienne.cavalier@chuliege.be (E.C.); aladang@chuliege.be (A.L.); 4Biochemistry Department, College of Science, King Saud University, Riyadh 11362, Saudi Arabia; jyreginster@uliege.be (J.-Y.R.); malokail@ksu.edu.sa (M.A.); 5Azienda USL Toscana Sud Est, 52100 Arezzo, Italy (Previously Responsible for the Clinic for the Diagnosis and Management of Hand Osteoarthritis, of the Rheumatology Unit, Department of Medicine, Surgery and Neuroscience, Azienda Ospedaliera Universitaria Senese, Policlinico Le Scotte, 53100 Siena, Italy);; 6Department of Medical and Surgical Sciences, Alma Mater Studiorum University of Bologna, 40126 Bologna, Italy; giorgiobedogni@unibo.it; 7Department of Primary Health Care, Internal Medicine Unit Addressed to Frailty and Aging, S. Maria delle Croci Hospital, AUSL Romagna, 48121 Ravenna, Italy

**Keywords:** erosive osteoarthritis of the hand, psoriatic arthritis, microRNAs, cytokines, miR-155, adipokines, biomarkers

## Abstract

The differential diagnosis of erosive osteoarthritis of the hand (EHOA) and psoriatic arthritis (PsA) is challenging, especially considering the absence of specific diagnostic biomarkers. The aim of the present study was to evaluate whether a pattern of microRNAs (miRNAs) (miR-21, miR-140, miR-146a, miR-155, miR-181a, miR-223), pro-inflammatory cytokines [interleukin (IL)-1β, IL-6, IL-17a, IL-23a, and tumor necrosis factor (TNF)-α], and adipokines (adiponectin, chemerin, leptin, resistin, and visfatin) could help to differentiate EHOA from PsA. Fifty patients with EHOA, fifty patients with PsA, and fifty healthy subjects (HS) were studied. The gene expression of miRNAs and cytokines were evaluated by real-time PCR from peripheral blood mononuclear cells and serum levels of cytokines and adipokines were quantified by ELISA in PsA and EHOA patients and HS. Gene expression showed the significant up-regulation of the analyzed miRNAs in EHOA and PsA patients as compared to HS and higher miR-155 in EHOA vs. PsA patients. The expression levels of IL-1β and IL-6 did not show any significant differences between EHOA and PsA, while IL-17a and IL-23a were significantly up-regulated in PsA compared to EHOA. Circulating TNF-α levels were higher in EHOA compared to PsA, while PsA patients exhibited significantly elevated levels of IL-23a. The combination of miR-155 with C-reactive protein enhanced the ability to differentiate EHOA from PsA, further supporting the potential of miR-155 as a diagnostic biomarker.

## 1. Introduction

The hand is a common site of osteoarthritis (OA) and carries a significant medical burden, causing persistent pain and functional limitations in everyday activities, leading to diminished quality of life [1,2,3]. According to the Global Burden of Disease Study, hand osteoarthritis (HOA) represented 24% of all cases of OA in 2019, with an incidence rising from 371 million cases in 1990 to 676 million cases in 2019 [4,5]; moreover, cases of HOA are projected to increase by nearly 50% by 2050 [6].

Erosive osteoarthritis of the hand (EHOA) is a rare subset of HOA that affects mainly postmenopausal middle-aged women. It is characterized by prominent signs of inflammation, severe progression, and peculiar radiographic changes in the interphalangeal (IP) joints [7,8,9,10]. It is currently debated whether EHOA is an advanced stage of classical HOA or a separate entity with typical inflammatory characteristics that can mimic chronic arthritis, such as psoriatic arthritis (PsA) [11,12].

PsA is a heterogeneous musculoskeletal inflammatory condition that can affect up to 30% of psoriasis patients [13]. It has various clinical features, including peripheral arthritis, spinal spondylitis, asymmetrical synovitis, enthesitis, and dactylitis [14,15]. Such heterogeneity and the absence of specific biomarkers make the diagnosis of PsA difficult. The CASPAR (Classification for Psoriatic Arthritis) criteria, which includes evidence of psoriasis (current, personal, or family history), dactylitis, nail dystrophy, a negative rheumatoid factor (RF), and radiographic evidence of new bone formation, is the current gold standard for diagnosing PsA [16].

Some features of PsA are found in other chronic musculoskeletal diseases, such as rheumatoid arthritis (RA) and EHOA, which makes it possible to delay the diagnosis and influence the success of treatment. The differential diagnosis of PsA and EHOA is especially challenging, considering that both conditions are characterized by bone proliferation and inflammation in the distal IP joints, and there are no diagnostic biomarkers [17].

The past decade has seen the emergence of microRNAs (miRNAs) as potential biomarkers for certain rheumatic diseases [18]. MiRNAs are small non-coding RNA molecules that control the expression of different target genes by repressing or inhibiting translation. Mature miRNAs are produced inside the cell and exert their function within the cytoplasm, but also by being released into the circulation and body fluids, where they regulate both physiological and pathological processes [19,20].

Specific miRNAs have been associated with the up-regulation of several inflammatory cytokines or degrading enzymes involved in the pathogenesis of PsA and OA [21,22]. Indeed, miRNAs have been detected in the plasma and synovial fluid of patients with PsA, and they are considered potential diagnostic and prognostic biomarkers [23,24,25,26]. Recently, Baloun et al. observed an elevated expression of a specific pattern of circulating miRNAs in patients with HOA compared to healthy individuals [27]. Auroux et al. further proposed that circulating miRNAs could serve as a diagnostic tool to distinguish EHOA from non-erosive HOA [28]. Notably, they identified an association between the down-regulation of miR-196a-5p and the presence of EHOA [28].

Adipokines, primarily produced by white adipose tissue, including adiponectin, leptin, visfatin, resistin, chemerin, and omentin, play a pivotal role in the pathogenesis of inflammatory and degenerative musculoskeletal disorders [29,30,31]. Additionally, adipokines are potential biomarkers and pharmacological targets in both PsA and HOA [32,33]. In a previous study, we found that circulating miR-140 and leptin levels were elevated in patients with PsA compared to those with RA [34]. Thus, miR-140 and leptin could serve as potential biomarkers for differentiating between these two inflammatory diseases.

The aim of the present study was to evaluate whether a pattern of miRNAs, adipokines, and pro-inflammatory cytokines could help to differentiate EHOA from PsA. In detail, we evaluated the expression profile of miR-21, miR-140, miR-146a, miR-155, miR-181a, and miR-223; interleukin (IL)-1β, IL-6, IL-17a, and IL-23a; and tumor necrosis factor (TNF)-α in peripheral blood mononuclear cells (PBMCs) of patients with EHOA, PsA, and healthy subjects (HS). Serum levels of IL-1β, IL-6, IL-17a, IL-23a, and TNF-α and adipokines (adiponectin, chemerin, leptin, resistin, and visfatin) were also investigated.

## 2. Results

### 2.1. Study Participants

Table 1 provides the main demographic and clinical features of the study subjects. All EHOA patients were negative for the rheumatoid factor (RF) and anti-cyclic citrullinated peptide antibodies (ACPAs); 6% of PsA patients had a positive RF in the absence of ACPA. Age was significantly lower in HS than in EHOA and PsA patients, as well as in PsA vs. EHOA patients. Low density lipoprotein (LDL) cholesterol was significantly higher in EHOA patients compared to HS and PsA patients. Erythrocyte sedimentation rate (ESR) and C reactive protein (CRP) were higher in PsA compared to HS and EHOA.

A significant difference in the Health Assessment Questionnaire (HAQ) score was detected between HS vs. EHOA patients and for HS vs. PsA patients.

### 2.2. MiRNAs, Cytokines, and Adipokines

Table 2 reports the gene expression of miRNAs and cytokines in PBMCs and the serum levels of cytokines and adipokines.

Gene expression showed a significant up-regulation of the analyzed miRNAs in EHOA and PsA patients as compared to HS. MiR-146a was significantly higher in PsA than in EHOA patients, while MiR-21 and MiR-155 were significantly up-regulated in EHOA vs. PsA patients. No significant differences were detected between EHOA and PsA patients for the expression profile of the remaining miRNAs.

The expression levels of IL-1β, IL-6, and TNF-α were similar in EHOA and PsA patients while IL-17a and IL-23a were significantly more up-regulated in PsA than in EHOA patients.

Serum levels of TNF-α were higher in EHOA than in PsA patients, while those of IL-23a were higher in PsA than in EHOA patients. Circulating levels of adiponectin, chemerin, leptin, resistin, and visfatin were significantly higher in PsA than in HS. EHOA patients had significantly higher values of adiponectin, chemerin, and resistin compared to HS. Leptin, resistin, and visfatin were significantly higher in PsA vs. EHOA patients.

Appendix A reports the association between miRNAs, cytokines, gene expression, and serum cytokines and adipokines, as detected by Spearman’s rank correlation coefficient.

### 2.3. Diagnostic Performance of miRNA and Cytokine Expression

The uni-variable logistic regression models used to evaluate the ability of miRNA and cytokine expression to discriminate EHOA from PsA are given in Table 3.

Among the gene expression profiles, miR-21, miR-146a, miR-155, TNF-α, IL-17a, and IL-23a were able to discriminate EHOA from PsA, with the corresponding probability curves plotted in Figure 1. miR-155 was the expression profile most strongly associated with EHOA vs. PsA (AIC = 89, BIC = 94, C-statistic = 0.89, and Nagelkerke R^2^ = 0.55), followed by IL-23a (AIC = 117, BIC = 122, C-statistic = 0.79, and Nagelkerke R^2^ = 0.30) and IL-17a (AIC = 128, BIC = 133, C-statistic = 0.74, and Nagelkerke R^2^ = 0.18).

The bi-variable logistic regression models used to evaluate the ability of miRNA and cytokine expression profiles to discriminate EHOA from PsA after correction for a known or potential confounder [sex, age, disease duration, body mass index (BMI), tender joints, and log-transformed CRP (lnCRP)] are given in Appendix A. As a rule, the association of the predictor with the outcome changed little after controlling for each confounder. Importantly, lnCRP alone was superior to most expression profiles at discriminating EHOA from PsA (Model M8 of Appendix A). Even if miR-155 was inferior to lnCRP according to all metrics, their combination greatly improved the discrimination of EHOA from PsA as compared to lnCRP alone (AIC = 58 vs. 78, BIC = 66 vs. 83, c-statistic = 0.95 vs. 0.90, Nagelkerke R^2^ = 0.77 vs. 0.64). The same pattern was observed for IL-23a, and to a lesser extent for IL-17a.

### 2.4. Diagnostic Performance of Serum Cytokines and Adipokines

The uni-variable logistic regression models used to evaluate the ability of serum cytokines and adipokines to discriminate EHOA from PsA are given in Table 4.

IL-6, TNF-α, IL-23a, chemerin, leptin, visfatin, and resistin were the cytokines and adipokines associated with the outcome, with the corresponding probability curves plotted in Figure 2. Among them, TNF-α had the strongest association with the outcome (AIC = 34, BIC = 39, C-statistic = 0.98, and Nagelkerke R^2^ = 0.88), followed by leptin (AIC = 81, BIC = 86, C-statistic = 0.91, and Nagelkerke R^2^ = 0.61) and visfatin (AIC = 86, BIC = 91, C-statistic = 0.89, and Nagelkerke R^2^ = 0.58

## 3. Discussion

Distinguishing between EHOA and PsA in the presence of peripheral arthritis can be challenging for physicians managing musculoskeletal disorders. Both diseases are, in fact, characterized by a prominent involvement of the distal IP joints of the hand and share similar clinical features, such as joint swelling, pain, tenderness, and the development of severe deformities with considerable functional limitations [9,12]. Moreover, the lack of specific autoantibodies and the inconsistent increase in acute-phase reactants in PsA and in EHOA make it more difficult to differentiate the two diseases [35,36]. Since the therapeutic options for PsA and EHOA are very different, a correct early diagnosis is essential for the success of the pharmacological treatments and to limit the progression of joint damage [12]. These considerations underscore the urgent need to identify novel, specific, and reliable biomarkers that can enhance diagnostic accuracy.

In this study, we investigated the potential of a panel of miRNAs and pro-inflammatory cytokines in PBMCs from patients with EHOA and PsA, as well as the serum levels of a specific pattern of cytokines and adipokines, to differentiate between the two diseases. In this study, we investigated the potential of a panel of miRNAs and pro-inflammatory cytokines in PBMCs from patients with EHOA and PsA, as well as the serum levels of a specific pattern of cytokines and adipokines, to differentiate between the two diseases. Increasing evidence supports the critical role of miRNAs in the pathogenesis of several musculoskeletal diseases, including PsA and OA. In recent years, circulating miRNAs have emerged as potential biomarkers for diagnosing and predicting the prognosis of various pathological conditions, including cardiovascular, neurological, oncological, metabolic, and rheumatological diseases. In fact, miRNAs are highly stable in biological fluids and resistant to endogenous ribonuclease activity, as well as to extreme conditions such as high temperatures, long-term storage, and repeated freeze–thaw cycles. Additionally, circulating miRNAs are easily accessible and can be measured with high sensitivity [21,26,34,37,38,39,40].

A recent study evaluated the profile of circulating miRNAs in 96 patients with EHOA, 73 patients with non-erosive HOA, and 69 HS [27]. Through a low-density array analysis of a large series of miRNAs, followed by a two-phase validation process, the authors discovered higher levels of miR-23a-3p, miR-146a-5p, and miR-652-3p in patients with non-erosive HOA and EHOA while no differences were detected between the two HOA groups [27]. Moreover, the same miRNAs were positively correlated with the Australian/Canadian Hand Osteoarthritis index (AUSCAN) sum score and with the AUSCAN score for pain. Interestingly, miR-222-3p exhibited an inverse correlation with the Kallman score [27]. Auroux et al. evaluated the circulating miRNA signature in patients with erosive and non-erosive HOA, showing a significant down-regulation of miR-196-5p in EHOA without any association with clinical symptoms [28].

The utility of miRNAs as biomarkers in PsA is less extensively explored compared to OA or other rheumatological diseases like RA or ankylosing spondylitis. The earliest clinical study that investigated the expression profile of a specific pattern of miRNAs in a PsA population was performed in 2017 [41]. The authors conducted a microarray analysis to identify differentially expressed miRNAs in the PBMCs of patients with PsA compared to HS. Among the identified miRNAs, miR-21-5p exhibited a significant up-regulation in PsA patients. This miRNA was believed to serve as a marker for treatment response, as its expression decreased after 12 weeks of treatment with methotrexate or etanercept and was correlated with a reduction in the Disease Activity for Psoriatic Arthritis (DAPSA) score [41]. A subsequent study analyzing the expression profile of miRNAs in patients with active or inactive PsA revealed distinct miRNA expression profiles for each of the two distinct clinical phases of the disease [42]. More recently, several studies identified other miRNAs (miR-21, miR-23a, miR-26a, miR-130, miR-140, miR-146a, miR-151, miR-155, miR-181-a, miR-221, miR-223, let7-e, and miR-6891-3p) as potential biomarkers to diagnose PsA and to monitor disease activity [23,25,26,43,44,45].

In a previous study, we explored the ability of a specific pattern of miRNAs (miR-21, miR-140, miR-146a, miR-155, miR-181b, miR-223, and miR-let-7e) in distinguishing PsA from RA [34]. Consistent with the current findings, we found a dysregulation of all the examined miRNAs in patients with PsA compared to HS. In the present study, we found a notable up-regulation in the gene expression of the considered miRNAs (miR-21, miR-140, miR-146a, miR-155, miR-181b, miR-223) in patients with PsA and EHOA compared to HS. Additionally, we observed a significant increase in the gene expression of miR-146a in patients with PsA compared to EHOA, while miR-155 and miR-21 showed a higher expression in EHOA compared to PsA.

In our experience, miR-155 stood out as the most effective miRNA in distinguishing EHOA from PsA. MiR-155 has garnered considerable attention due to its involvement in various pathological conditions. Dysregulation of miR-155 has been reported in many inflammatory autoimmune diseases, including RA, systemic lupus erythematosus, multiple sclerosis, Sjögren’s syndrome, systemic sclerosis, and inflammatory bowel disease [22,46]. Indeed, various studies have demonstrated that miR-155 is one of the most up-regulated miRNAs in human OA cartilage and plays a crucial role in the pathogenesis of its disease by regulating essential cellular mechanisms, including the proliferation, apoptosis, pyroptosis, differentiation, growth, and migration of chondrocytes [47,48]. In OA, miR-155 was shown to act through different signaling cascades, such as the mitogen-activated protein kinase (MAPK) pathway, to regulate chondrocyte proliferation and apoptosis and contribute to extracellular matrix degradation [49].

Clinical data on circulating levels of miR-155 in OA are limited to a few studies in gonarthrosis. Our study is the first to explore the gene expression of miR-155 in EOHA and our findings align with previous reports. Okuhara et al. [50] and Soyocak et al. [51] reported an increased expression of miR-155 in the PBMCs of patients with knee OA compared to healthy subjects. Both research groups also observed a correlation between miR-155 levels and the later stages of the disease. Giannitti et al. further demonstrated that miR-155 could serve as a reliable marker for therapeutic response in patients with knee OA treated with mud bath therapy [52]. Taken together, these data suggest the broad relevance of miR-155 in the development and progression of osteoarthritis (OA) not only in the hand but also in other joints. The up-regulation of miR-155 may partially explain our findings of the high gene expression and elevated serum levels of cytokines IL-1β, IL-6, and TNF-α in EHOA. As reported by O’Connell et al., this miRNA stimulates dendritic cells to produce specific cytokines essential for the development of inflammatory T cells [53]. Further research confirms that the up-regulation of miR-155 led to the production of pro-inflammatory cytokines by targeting Src homology 2-containing inositol phosphatase-1 (SHIP-1), an inhibitor of inflammation [54,55].

Moreover, we demonstrated for the first time that miR-21 is up-regulated in patients with EHOA compared to HS and PsA patients. MiR-21 plays a crucial role in inflammation and may represent a potential therapeutic target in OA [56,57]. Wang et al. found a substantial increase in miR-21 levels in OA cartilage compared to cartilage from patients who had experienced traumatic events without a history of OA [58]. Mir-21, abundant in synovial tissue and fluid, may contribute to knee OA pain by activating Toll-like receptor (TLR) 7 [59,60].

In agreement with the previous literature, we confirmed the up-regulation of miR-146a in PsA patients compared to HS and its potential role as a biomarker of the disease [23,26,41]. Moreover, our findings suggest that miR-146a may be useful in discriminating EHOA from PSA.

Studies exploring gene expression and/or serum levels of inflammatory cytokines in EHOA have yielded conflicting results, likely due to the diverse clinical characteristics of the analyzed patients [61,62]. Recently, McAlindon et al. observed a strong and consistent association between elevated serum levels of IL-7, a cytokine that induces inflammation, cartilage destruction, and bone loss, and incident radiographic evidence of EHOA [63].

Surprisingly, our study found higher serum levels of TNF-α in EHOA compared to PsA. Moreover, TNF-α could effectively discriminate EHOA from PsA. It is challenging to explain these results, even though EHOA is characterized by clinical and radiographic inflammatory hallmarks [64]. Regrettably, we did not conduct any ultrasound or magnetic resonance imaging studies to assess the presence and severity of inflammation. Therefore, we cannot entirely rule out the possibility that our results may be partially influenced by the relatively low inflammatory activity in patients with PsA. Our findings regarding serum levels of TNF-α in EHOA warrant further investigation in larger studies and with appropriate imaging assessments. If confirmed, these findings could pave the way for novel therapeutic perspectives, particularly for a targeted (precision) treatment of EHOA. Unfortunately, most of the currently available conventional and biological disease-modifying anti-rheumatic drugs have failed to yield significant clinical benefits [9,65].

In line with the existing literature, our study underscores the crucial role of IL-17a and IL-23a in the pathogenesis of PsA, as well as their potential as biomarkers and therapeutic targets for the disease [66].

In the present study, we also analyzed a panel of adipokines and observed higher levels of adiponectin, chemerin, leptin, resistin, and visfatin in patients with PsA compared to HS. The literature on circulating adipokines in PsA and EHOA is limited and frequently contradictory [30,32,63,67]. This may be due to differences in patient demographic and clinical features. Several factors, such as age, gender, smoking habits, disease duration, radiographic features, obesity, metabolic syndrome, and concomitant comorbidities, are indeed known to affect circulating adipokines [68,69].

In the present study, lnCRP had greater discriminatory power compared to all miRNAs in distinguishing EHOA from PsA. This finding aligns with a substantial body of the literature, as CRP is a well-established non-specific marker of active inflammation. CRP has been suggested as a laboratory marker for PsA and is included in the most used response criteria for PsA, such as the ACR response and the Disease Activity Score 28 with CRP (DAS28-CRP) [70]. However, the association of CRP with EHOA remains a subject of debate. Some studies have reported higher CRP levels in EHOA compared to its non-erosive counterpart, while others have observed no significant differences, or even opposite results [71,72]. We found that the combination of miR-155 with ln-CRP enhanced the discriminatory ability of EHOA from PsA.

Overall, the present study supported the role of MiRNAs as promising biomarkers for distinguishing EHOA from PsA. However, their practical application is still limited by several factors, including the absence of a standardized protocol for detecting and quantifying circulating miRNAs. This could explain the significant variability and heterogeneity of the results reported in the literature [18,24]. Furthermore, incorporating miRNA biomarkers into routine practice necessitates an evaluation of their feasibility and cost-effectiveness compared to other laboratory tests or imaging techniques.

The present study had some limitations. Firstly, the sample size was relatively small, owing to the stringent inclusion criteria and the need to minimize any potential interference from pharmacological agents on miRNAs, cytokines, and adipokines. Secondly, the absence of patients with PsA without psoriasis, i.e., “sine psoriasis”, and the lack of comparison with a group exclusively affected by cutaneous lesions, further limited the study’s generalizability. Thirdly, despite our best efforts to match patients and controls, the median age differed among the study groups. Lastly, the case–control design, while an effective approach for generating hypotheses, required further validation through cross-sectional and possibly cohort studies where the diagnosis is not yet known [73].

In conclusion, we identified miR-155 as a biomarker capable of distinguishing EHOA from PsA. When combined with ln-CRP, its diagnostic performance is improved. However, external validation of our findings is necessary to confirm that these biomarkers offer a simple and reliable tool for accurate diagnosis, particularly in the early stages of the disease.

## 4. Materials and Methods

### 4.1. Study Design

From September 2018 to December 2021, a case–control study was conducted on outpatients seen at the Center for the Diagnosis and Management of Hand Osteoarthritis, Rheumatology Unit of the Azienda Ospedaliera Universitaria Senese (Siena-Italy). All participants gave written informed consent, and their records were anonymized before data analysis. The study was approved by the Local Ethical Committee (decision number 14047-2018) and was conducted in accordance with the Declaration of Helsinki. The study was registered on http://www.clinicaltrials.gov with number NCT05683769 (accessed on 1 September 2018).

### 4.2. Study Participants

The study involved 50 patients with EHOA, 50 with PsA, and 50 HS. The patients with PsA and the HS had been studied before [34]. As in our previous study [34], the HS were analyzed to obtain an idea of the values of the biomarkers in a healthy population. The case–control study was conducted to determine whether PsA can be distinguished from EHOA by utilizing such biomarkers. To accomplish this objective, “cases” were defined as patients with EHOA and “controls” as those with PsA.

The inclusion criteria for PsA patients and for HS were reported in our previous study [34]. In detail, patients with PsA who met the CASPAR classification criteria and exhibited a peripheral arthritis pattern were included in the study [16]. All the subjects had at least a moderate degree of disease activity defined by DAPSA and received a diagnosis of psoriasis. They were naïve to conventional and biologic diseases-modifying anti-rheumatic drugs and were only undergoing topical treatment for cutaneous lesions. The diagnosis of EHOA was based on the American College of Rheumatology criteria for HOA [74] and the presence of classical central erosion in at least two IP joints [7,75] in a blind observation. Disagreements were resolved by a third examiner (J-Y.R.). The Kallman score assessed osteophytes (0–3) and lateral deformities (0–1) in 20 joints; joint space narrowing (0–3), subchondral sclerosis (0–1), and subchondral cysts (0–1) in 22 joints; and erosions (0–1) in 18 joints, resulting in a score ranging from 0 to 198 [76].

Exclusion criteria were inflammatory rheumatic and bowel diseases, autoimmune and endocrine disorders, and other rheumatic and non-rheumatic conditions that could affect the functionality of the peripheral joints, such as tendinopathies, carpal tunnel syndrome, Dupuytren’s contracture, collagen and neurological disorders, or arthroplasty of the upper and lower limbs. Other exclusion criteria were diabetes mellitus, liver and kidney diseases, acute or chronic infectious disorders, cancer within the past 5 years, pregnancy and breastfeeding, a BMI > 30 kg/m^2^, the use of anti-obesity medications, and recent trauma or surgery to the affected joints. Patients with pure or mixed axial involvement were also excluded. Additionally, patients who had received systemic or intra-articular (i.a.) corticosteroids or i.a. hyaluronic acid within the past 3 months, or intra-muscular or intra-venous bisphosphonates within the past 6 months, were also excluded [77].

### 4.3. Clinical Examination

The clinical examination was performed by two expert rheumatologists (AF, ST). Weight and height were measured following international guidelines [78]. BMI was calculated as weight (kg)/height (m)^2^ and classified according to the National Institutes of Health [79].

### 4.4. Questionnaires and Scales

The global joint pain was calculated using a 0 to 100 mm visual analog scale (VAS), with 0 representing the absence of pain and 100 the maximum imaginable pain. The general health status was measured using the Italian version of the HAQ. HAQ is a self-administered questionnaire developed to measure disability and consists of 8 sections: dressing, arising, eating, walking, hygiene, reach, grip, and activities, and ranges from 0 to 3, with 3 corresponding to the highest level of disability [80,81]. For EHOA patients, the functional disability of the hand was assessed using the Italian version of the functional index for hand osteoarthritis (FIHOA). The score ranges from 0 to 30, with higher scores indicating the most severe functional impairment [82]. Moreover, the Kallman score was used for radiological evaluation [76]. The disease activity of PsA patients was evaluated using the DAPSA score. The cut points for low and high disease activity were set at 18.5 and 45.1, respectively [83]. The severity and extent of skin psoriasis in PsA patients was measured using the Psoriasis Area Severity Index (PASI), which has a maximum score of 72 [84].

### 4.5. Laboratory Analysis

Blood samples were drawn from an antecubital vein in the morning after an overnight fast. A portion of whole blood was immediately centrifuged to evaluate ESR, CRP, total cholesterol, HDL cholesterol, LDL cholesterol, triglycerides, and glucose. IgG, IgA, and IgM RF were assessed using commercial ELISA kits (Orgentec Diagnostica, Mainz, Germany). ACPAs were measured using the FEIA technique with the EliA system (Phadia Diagnostics, Freiburg, Germany) [35].

Serum samples were stored at −80 °C for the measurement of cytokines and adipokines using ELISA.

The isolation of PBMCs from whole blood was performed using the standard classical method of Ficoll density gradient centrifugation (Ficoll-Paque GE HealthCare, Little Chalfont, Buckinghamshire, UK), as per the manufacturer’s instructions [34,85,86].

### 4.6. MiRNA and Cytokine Expression Analysis

Total RNA was extracted using the TriPure Isolation Reagent (Euroclone, Milan, Italy), following the manufacturer’s instructions. The concentration, purity, and integrity of the RNA were determined using a Nanodrop-1000 (Celbio, Milan, Italy), and their quality was verified by electrophoresis on an agarose gel (FlashGel System, Lonza, Rockland, ME, USA).

A total of 500 ng of RNA were reverse-transcribed into cDNA using a commercial kit for miRNAs (Qiagen, Hilden, Germany), as per the manufacturer’s instructions. The obtained cDNA was processed through real-time PCR using a kit that utilized the SYBR Green assay (Qiagen, Hilden, Germany). All PCR reactions were prepared in glass capillaries and analyzed using a LightCycler 1.0 instrument (Roche Molecular Biochemicals, Mannheim, Germany) coupled with LightCycler software Version 3.5. The list of primers employed for the PCR reactions is provided in Appendix A.

To verify the correct amplification of the PCR products, the analysis of the dissociation curves was performed to visualize the amplicon lengths in agarose gel. The Ct values and efficiency of the primer set were evaluated and converted into the relative expression (RE) [87,88]. Data normalization was assessed using Small Nucleolar RNA, C/D Box 25 (SNORD-25) as a housekeeping gene for miRNAs, and Actin Beta (ACTB) as a housekeeping gene for target genes [34,89].

### 4.7. Serum Cytokines and Adipokines

The methods for the of measurement of cytokines and adipokines have been reported in detail elsewhere [34].

Serum levels of IL-1β, IL-6, and TNF-α were determined using a Human Picokine ELISA kit (Boster Biological Technology, Pleasanton, CA, USA). The IL-1β kit had a sensitivity of 0.15 pg/mL, with inter- and intra-assay coefficients of variation (CVs) ranging from 5.7% to 8.9%. The IL-6 kit had a sensitivity of 0.3 pg/mL, with inter- and intra-assay CVs of 7.2% to 8.6%. The TNF-α kit had a sensitivity of 0.1 pg/mL, with inter- and intra-assay CVs of 5.4% to 6.4%.

Serum IL-17a was measured using the Cymax Human IL-17a ELISA kit (AbFRONTIER, Vinci-Biochem, Firenze, Italy), with a sensitivity of 2.134 pg/mL and inter- and intra-assay CVs of 4.42% to 6.35%.

Serum IL-23a was measured using the Human IL-23 ELISA Kit ab64708 (Abcam, Milan, Italy), with a sensitivity of 20 pg/mL and a detection range of 156.2 to 5000 pg/mL.

Serum adiponectin was measured using the Human Adiponectin ELISA kit (AdipoGen Life Sciences, Liestal, Switzerland), with a sensitivity of 100 pg/mL and inter- and intra-assay CVs of 2.8% to 5.5%.

Serum chemerin was measured using the Human Picokine ELISA kit (Boster Biological Technology, Pleasanton, CA, USA), with a sensitivity of 20 pg/mL and inter- and intra-assay CVs of 6.0% to 9.3% [34].

Serum leptin was measured using the Human Picokine ELISA kit (Boster Biological Technology, Pleasanton, CA, USA), with a sensitivity of 10 pg/mL and inter- and intra-assay CVs of 7.0–8.4% and 5.2–7.6%, respectively.

Serum resistin was measured using the Human Resistin ELISA kit (AdipoGen Life Sciences, Liestal, Switzerland), with a sensitivity of 3 pg/mL and inter- and intra-assay CVs of 4.20–7.20% and 2.86–5.17%, respectively [34].

Lastly, serum visfatin was measured using the Human Nampt (Visfatin/PBEF) ELISA kit (AdipoGen Life Sciences, Liestal, Switzerland), with a sensitivity of 30 pg/mL and inter- and intra-assay CVs of 4.66–7.40% and 2.31–9.11%, respectively.

### 4.8. Statistical Analysis

Most continuous variables were not Gaussian-distributed, and all are reported as the median (50th percentile) and interquartile interval (IQI, 25th and 75th percentile). Discrete variables are reported as the number and proportion of subjects with the characteristic of interest.

Between-group comparisons of continuous variables were performed using the Mann–Whitney *U*-test and those of discrete variables using Pearson’s Chi-square test.

The association between miRNAs, cytokines, and adipokines was evaluated using Spearman’s rank correlation coefficient.

CRP was transformed by adding 0.01 to its value, which ranged from 0 to 2.9, and then log-transformed by taking the natural logarithm (ln) of the resultant value. This transformation reduced the skewness of CRP and allowed for the assumptions made by the uni-variable and bi-variable logistic regression models which used it as a predictor to be met.

Uni-variable logistic regression models were used to evaluate the ability of continuous miRNAs to discriminate EHOA from PsA [90]. Six prespecified bi-variable logistic regression models were used to evaluate the ability of each miRNA to discriminate EHOA from PsA after correction for a potential predictor, i.e., sex (discrete: 0 = no; 1 = yes), age (continuous, years), disease duration (continuous, months), BMI (continuous, kg/m^2^), tender joints (continuous), and ln-CRP (continuous, U/L) [34,90]. We evaluated the linearity of the logits using scatterplots and uni- and multi-variable fractional polynomials [34,91]. There was no or a very modest increase in model fit using logit transformations, so that logits were kept linear and were modeled as such.

We compared the logistic regression models using the Akaike information criterion (AIC) and the Bayesian information criterion (BIC) and additionally calculated the Nagelkerke pseudo-R^2^ and Harrell’s C-statistic. Statistical analysis was performed using Stata 18.5 (Stata Corporation, College Station, TX, USA).

## Figures and Tables

**Figure 1 ijms-26-04621-f001:**
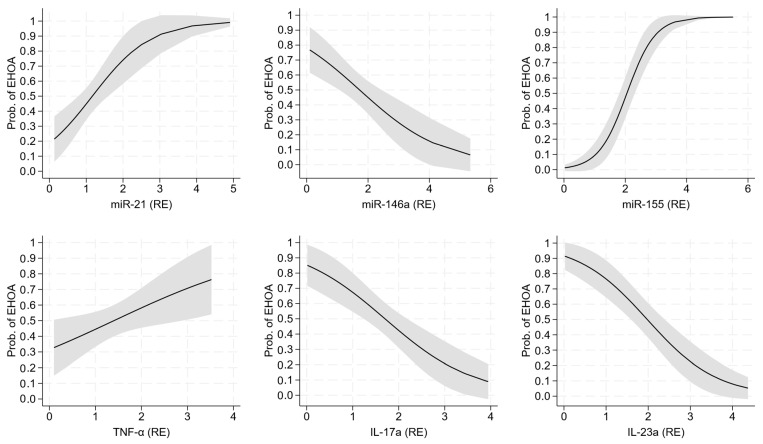
Probability of erosive osteoarthritis of the hand vs. psoriatic arthritis according to the expression levels of miR-21, miR-146a, miR-155, TNF-α, IL-17a, and IL-23a. The underlying logistic regression models are given in Table 3. Legend: EHOA = erosive osteoarthritis of the hand; RE = relative expression.

**Figure 2 ijms-26-04621-f002:**
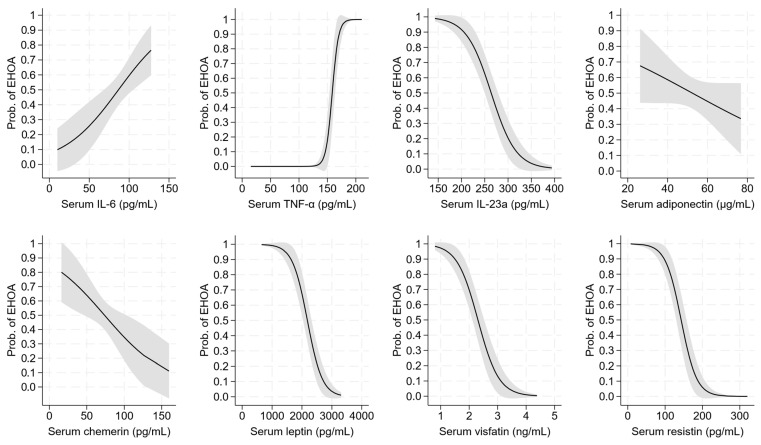
Probability of erosive osteoarthritis of the hand vs. psoriatic arthritis according to the serum levels of IL-6, TNF-α, IL-23a, adiponectin, chemerin, leptin, visfatin, and resistin. The underlying logistic regression models are given in Table 4. Legend: EHOA = erosive osteoarthritis of the hand.

**Table 1 ijms-26-04621-t001:** Demographic and clinical features of the study subjects.

	HS (*n* = 50)	EHOA (*n* = 50)	PsA (*n* = 50)	*p*-Value * HS vs. EHOA	*p*-Value * EHOA vs. PsA	*p*-Value * HS vs. PsA
Female sex	31 (62%)	40 (80%)	28 (56%)	0.047	0.010	0.54
Age (years)	48 (40;59)	68 (65;73)	58 (55;63)	<0.001	<0.001	<0.001
Disease (months)	—	96 (60;120)	72 (48;96)	—	0.014	—
BMI (kg/m^2^)	23.8 (21.9;25.2)	24.8 (22.5;26.3)	25.0 (23.2;26.8)	0.042	0.45	0.004
Smoking	13 (26%)	4 (8%)	19 (38%)	0.017	<0.001	0.20
Hypertension	12 (24%)	24 (48%)	21 (42%)	0.012	0.55	0.056
Cardiovascular disease	10 (20%)	9 (18%)	18 (36%)	0.80	0.043	0.075
Type 2 diabetes mellitus	0 (0%)	7 (14%)	12 (24%)	0.006	0.20	<0.001
Glucose (mg/dL)	87 (78;95)	95 (88;101)	90 (85;97)	0.001	0.016	0.15
Total cholesterol (mg/dL)	180 (165;195)	210 (183;233)	182 (175;195)	<0.001	<0.001	0.38
HDL cholesterol (mg/dL)	60 (52;65)	60 (51;69)	50 (42;57)	0.84	<0.001	<0.001
LDL cholesterol	99 (85;111)	140 (117;151)	105 (98;120)	<0.001	<0.001	0.005
Triglycerides (mg/dL)	110 (88;134)	130 (91;156)	120 (98;138)	0.047	0.41	0.14
ESR (mm/h)	12 (8;18)	15 (9;24)	35 (25;42)	0.062	<0.001	<0.001
CRP (mg/dL)	0.1 (0.0;0.1)	0.2 (0.1;0.5)	0.9 (0.7;1.4)	<0.001	<0.001	<0.001
VAS pain (0–100)	—	50 (20;70)	38 (25;60)	—	0.079	
HAQ	0 (0;0)	1 (0;1)	1 (0;1)	<0.001	0.41	<0.001
Tender joints (number)	—	7 (5;8)	8 (4;12)	—	0.12	—
Swollen joints (number)	—	4 (2;6)	2 (1;3)	—	<0.001	—
Kallman score	—	122 (98;150)	—	—	—	—
DAPSA-CRP	—	—	18 (14;29)	—	—	—
DAS28-ESR	—	—	5 (4;6)	—	—	—
PASI	—	—	6 (4;10)	—	—	—

* Between-group comparisons performed with Mann–Whitney *U*-test for continuous variables and with Pearson’s Chi-square test for categorical variables. Values are given as median and interquartile interval for continuous variables and as number and proportion for discrete variables. Legend: HS = healthy subjects; EHOA = erosive osteoarthritis of the hand; PsA = psoriatic arthritis; BMI = body mass index; ESR = erythrocyte sedimentation rate; CRP = C-reactive protein; VAS = visual assessment scale; HAQ = Health Assessment Questionnaire; DAPSA = Disease Activity in Psoriatic Arthritis; DAS28 = Disease Activity Score 28; PASI = Psoriasis Area Severity Index.

**Table 2 ijms-26-04621-t002:** MiRNA and cytokine expression and serum cytokines and adipokines of the study subjects.

	HS (*n* = 50)	EHOA (*n* = 50)	PsA (*n* = 50)	*p*-Value HS vs. EHOA *	*p*-Value EHOA vs. PsA *	*p*-Value HS vs. PsA *
miR-21 (RE)	0.43 (0.27;0.81)	1.19 (0.89;1.84)	0.92 (0.57;1.28)	<0.001	0.001	<0.001
miR-140 (RE)	0.73 (0.48;1.12)	1.75 (0.94;2.04)	1.55 (0.95;2.10)	<0.001	0.58	<0.001
miR-146a (RE)	0.74 (0.28;0.93)	1.13 (0.75;1.98)	1.86 (1.50;2.54)	<0.001	<0.001	<0.001
miR-155 (RE)	0.56 (0.30;0.75)	2.76 (2.12;3.23)	1.39 (0.92;1.84)	<0.001	<0.001	<0.001
miR-181b (RE)	0.80 (0.70;0.95)	3.43 (2.54;4.07)	2.97 (2.05;4.18)	<0.001	0.11	<0.001
miR-223 (RE)	1.19 (0.82;1.48)	2.30 (1.68;2.89)	2.44 (1.82;3.04)	<0.001	0.67	<0.001
IL-1β (RE)	0.82 (0.42;0.98)	1.03 (0.83;1.84)	1.31 (0.92;1.76)	<0.001	0.59	<0.001
IL-6 (RE)	0.31 (0.22;0.52)	1.10 (0.66;2.03)	1.58 (0.93;2.02)	<0.001	0.32	<0.001
IL-17a (RE)	0.27 (0.15;0.41)	1.19 (0.89;1.93)	2.03 (1.46;2.40)	<0.001	<0.001	<0.001
IL-23a (RE)	0.30 (0.18;0.52)	1.21 (0.90;2.04)	2.31 (1.80;3.24)	<0.001	<0.001	<0.001
TNF-α (RE)	0.57 (0.29;0.94)	1.49 (0.90;2.02)	1.03 (0.80;1.79)	<0.001	0.047	<0.001
IL1-β (pg/mL)	16.6 (12.1;23.1)	22.2 (18.4;26.9)	21.9 (16.9;27.1)	<0.001	0.43	0.002
IL-6 (pg/mL)	23.4 (17.2;30.7)	94.7 (71.1;110.3)	79.6 (66.1;100.1)	<0.001	0.004	<0.001
IL-17a (pg/mL)	46.3 (37.6;55.3)	70.0 (58.4;79.2)	71.8 (59.7;81.9)	<0.001	0.44	<0.001
IL-23a (pg/mL)	188.5 (171.0;201.0)	230.5 (195.0;266.0)	308.0 (256.0;338.0)	<0.001	<0.001	<0.001
TNF-α (pg/mL)	26.4 (20.8;31.8)	187.8 (172.3;196.9)	130.3 (122.9;141.9)	<0.001	<0.001	<0.001
Adiponectin (μg/mL)	43.73 (36.13;50.54)	49.49 (43.81;58.67)	51.33 (44.58;61.17)	<0.001	0.30	<0.001
Chemerin (pg/mL)	34.82 (29.29;44.55)	61.40 (54.63;83.34)	74.44 (67.44;85.96)	<0.001	0.002	<0.001
Leptin (pg/mL)	1716.12 (1492.12;1834.22)	1737.39 (1260.28;2033.62)	2637.39 (2282.95;2836.6)	0.35	<0.001	<0.001
Resistin (pg/mL)	74.87 (58.14;89.13)	95.64 (79.61;112.37)	198.08 (174.09;230.83)	<0.001	<0.001	<0.001
Visfatin (ng/mL)	1.88 (1.43;2.38)	1.78 (1.60;2.11)	2.95 (2.30;3.56)	0.35	<0.001	<0.001

* Between-group comparisons performed by Mann–Whitney *U*-test. Values are given as median and interquartile interval for continuous variables. Legend: HS = healthy subjects; EHOA = erosive osteoarthritis of the hand; PsA = psoriatic arthritis; RE = relative expression.

**Table 3 ijms-26-04621-t003:** Ability of miRNA and cytokine expression to discriminate EHOA from PsA.

	M1	M2	M3	M4	M5	M6	M7	M8	M9	M10	M11
miR-21 (RE)	1.26 ** [0.46, 2.06]	—	—	—	—	—	—	—	—	—	—
miR-140 (RE)	—	0.23 [−0.36, 0.81]	—	—	—	—	—	—	—	—	—
miR-146a (RE)	—	—	−0.74 ** [−1.21, −0.26]	—	—	—	—	—	—	—	—
miR-155 (RE)	—	—	—	2.16 *** [1.35, 2.97]	—	—	—	—	—	—	—
miR-181b (RE)	—	—	—	—	0.27 [−0.10, 0.63]	—	—	—	—	—	—
miR-223 (RE)	—	—	—	—	—	−0.11 [−0.60, 0.37]	—	—	—	—	—
IL-1β (RE)	—	—	—	—	—	—	−0.14 [−0.77, 0.49]	—	—	—	—
IL-6 (RE)	—	—	—	—	—	—	—	−0.11 [−0.59, 0.36]	—	—	—
TNF-α (RE)	—	—	—	—	—	—	—	—	0.55 * [0.01, 1.10]	—	—
IL-17a (RE)	—	—	—	—	—	—	—	—	—	−1.03 *** [−1.62, −0.45]	—
IL-23a (RE)	—	—	—	—	—	—	—	—	—	—	−1.21 *** [−1.76, −0.66]
Intercept	−1.47 ** [−2.46, −0.48]	−0.35 [−1.34, 0.64]	1.27 ** [0.38, 2.16]	−4.44 *** [−6.15, −2.73]	−0.85 [−2.09, 0.38]	0.27 [−0.94, 1.47]	0.20 [−0.75, 1.14]	0.16 [−0.61, 0.93]	−0.77 [−1.63, 0.08]	1.76 ** [0.69, 2.83]	2.39 *** [1.24, 3.54]
*N*	100	100	100	100	100	100	100	100	100	100	100
AIC	130	142	131	89	141	142	142	142	138	128	117
BIC	135	147	137	94	146	148	148	148	144	133	122
C-statistic	0.69	0.53	0.72	0.89	0.59	0.52	0.53	0.56	0.62	0.74	0.79
R^2^	0.16	0.01	0.14	0.55	0.03	0.00	0.00	0.00	0.05	0.18	0.30

* *p* < 0.05, ** *p* < 0.01, *** *p* <0.001. Values are coefficients from logistic regression with 95% confidence intervals in brackets. Legend: EHOA = erosive osteoarthritis of the hand; PsA = psoriatic arthritis; AIC = Akaike information criterion; BIC = Bayesian information criterion; RE = relative expression; R^2^ = Nagelkerke R^2^.

**Table 4 ijms-26-04621-t004:** Ability of serum cytokines and adipokines to discriminate EHOA from PsA.

	M12	M13	M14	M15	M16	M17	M18	M19	M20	M21
IL1-β (pg/mL)	0.01 [−0.05, 0.07]									
IL-6 (pg/mL)		0.03 ** [0.01, 0.05]								
TNF-α (pg/mL)			0.18 *** [0.09, 0.27]							
IL-17a (pg/mL)				−0.01 [−0.04, 0.01]						
IL-23a (pg/mL)					−0.04 *** [−0.05, −0.02]					
Adiponectin (μg/mL)						−0.03 [−0.07, 0.01]				
Chemerin (pg/mL)							−0.02 * [−0.05, −0.01]			
Leptin (pg/mL)								−0.004 *** [−0.006, −0.003]		
Visfatin (ng/mL)									−2.72 *** [−3.77, −1.68]	
Resistin (pg/mL)										−0.05 *** [−0.07, −0.03]
Intercept	−0.25 [−1.61, 1.11]	−2.50 ** [−4.31, −0.70]	−28.72 *** [−42.49, −14.95]	0.98 [−1.04, 3.00]	9.90 *** [6.07, 13.72]	1.47 [−0.59, 3.52]	1.78 * [0.13, 3.43]	8.78 *** [5.39, 12.17]	6.29 *** [3.90, 8.68]	6.91 *** [4.47, 9.34]
N	100	100	100	100	100	100	100	100	100	100
AIC	142	134	34	142	92	141	137	81	86	63
BIC	148	139	39	147	97	146	143	86	91	68
C-statistic	0.55	0.67	0.98	0.54	0.87	0.56	0.68	0.91	0.89	0.95
Nagelkerke R^2^	0.00	0.11	0.88	0.01	0.53	0.03	0.07	0.61	0.58	0.73

* *p* < 0.05, ** *p* < 0.01, *** *p* < 0.001. Values are coefficients from logistic regression with 95% confidence intervals in brackets. Legend: EHOA = erosive osteoarthritis of the hand; PsA = psoriatic arthritis.

## Data Availability

The data and the code used in the current study are available upon reasonable request from the corresponding author.

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
