# Peer review of "Can Circulating MicroRNAs, Cytokines, and Adipokines Help to Differentiate Psoriatic Arthritis from Erosive Osteoarthritis of the Hand? A Case–Control Study"

_ijms, 2025, doi:10.3390/ijms26104621_

Round 1

Reviewer 1 Report

Comments and Suggestions for Authors

>>Review for the paper titled:

 Can circulating MicroRNAs, Cytokines and Adipokines help to differentiate Psoriatic 2 Arthritis from Erosive Osteoarthritis of the Hand? A Case-Control Study.

>>Summary: The study aimed to evaluate whether patterns of microRNAs (miRNAs), pro-inflammatory cytokines, and adipokines can help differentiate osteoarthritis of the hand (EHOA) from psoriatic arthritis (PsA). The paper is innovative and well thought out.

>>Drawbacks: I have some minimal suggestions:

- In Table 2, does the abbreviation (RE) refer to the relative expression in arbitrary units?

-The discussion focuses on miR-155 due to the authors' experience, but it is not clear why miR-146 and miR-21 are not considered interesting as markers. This could be clarified.

-A representative agarose gel can be added to the supplementary material.

- "I checked the paper on the Turnitin platform and found a similarity of 44%, particularly in the 'Materials and Methods' section. To lower this percentage, I believe it is necessary to edit this part.

Author Response

General response to Reviewer #1

Many thanks for your constructive comments and valuable suggestions regarding our manuscript “Can circulating MicroRNAs, Cytokines, and Adipokines help differentiate Psoriatic Arthritis from Erosive Osteoarthritis of the Hand? A Case-Control Study.” These suggestions have significantly improved our paper. We addressed your concerns and hope that you will now find the revised article more suitable for publication in the IJMS. Below, you will find a point-by-point answer to your comments. All changes have been highlighted in red color in the revised manuscript.

Detailed responses to Reviewer #1

Summary - The study aimed to evaluate whether patterns of microRNAs (miRNAs), pro-inflammatory cytokines, and adipokines can help differentiate osteoarthritis of the hand (EHOA) from psoriatic arthritis (PsA). The paper is innovative and well thought out. Drawbacks: I have some minimal suggestions:

Q1: In Table 2, does the abbreviation (RE) refer to the relative expression in arbitrary units?

A1: Yes, this is now explained in the legends of tables and figure.

Q2: The discussion focuses on miR-155 due to the authors' experience, but it is not clear why miR-146 and miR-21 are not considered interesting as markers. This could be clarified.

A2: We sincerely thank the Reviewer for the constructive and helpful suggestion. According to  your recommendation we added new sentences with the appropriate references  in the Discussion Section  

Q3: A representative agarose gel can be added to the supplementary material.

A3: Although we fully comprehend the request, our laboratory was closed after the retirement of the first author, making it literally impossible to retrieve such data. We can only sincerely apologize for this.

Q 4: - "I checked the paper on the Turnitin platform and found a similarity of 44%, particularly in the 'Materials and Methods' section. To lower this percentage, I believe it is necessary to edit this part.

Reviewer 2 Report

Comments and Suggestions for Authors

  1. The study does not specify the number of patients, raising concerns about statistical data
  2. Findings should be validated in an independent cohort to confirm biomarker reliability.
  3. Did the study rule out RA or other arthritis types that could confound miRNA/cytokine profiles?
  4. Peripheral blood mononuclear cells (PBMCs) include mixed cell populations; cell-specific miRNA changes may be diluted.
  5. ELISA measures total protein levels but may miss post-translational modifications affecting cytokine activity.
  6. The study shows associations but does not demonstrate if miRNAs/cytokines play a causal role in disease mechanisms.
  7. Were comorbidities (obesity, diabetes) or medications accounted for, as they influence adipokines/cytokines?
  8. The discriminatory power of miR-155 + CRP needs AUC/ROC analysis to assess clinical utility.
  9. Were there PsA/EHOA patients with similar biomarker profiles, reducing diagnostic accuracy?
  10. Do these biomarkers change with disease progression or treatment?
  11. Higher TNF-α in EHOA contradicts its known role in PsA; is this due to differences in disease activity?
  12. Extracellular miRNAs can degrade; were pre-analytical conditions strictly controlled?
  13. miRNA qPCR data require proper reference genes; were stable controls used?
  14. Were corrections (e.g., Bonferroni) applied given the number of miRNAs/cytokines tested?
  15. Blood biomarkers may not reflect local joint pathology; were synovial samples analyzed?
  16. Adipokines (e.g., leptin) fluctuate with metabolic state; were fasting samples collected?
  17. miRNA profiling + ELISA may be expensive for routine diagnostics compared to imaging/clinical criteria.
  18. Lab-to-lab variability in miRNA quantification could limit widespread adoption.
  19. How do miR-155 and TNF-α interact in EHOA vs. PsA pathogenesis?
  20. Even if biomarkers differ, does this guide treatment decisions (e.g., anti-TNF for EHOA)?

Suggested Improvements

Include a larger, multi-center cohort,

Compare biomarkers to imaging/histopathology findings.

Perform functional assays (e.g., miRNA inhibition in cell models).

Author Response

General response to Reviewer #2

Many thanks for your constructive comments and valuable suggestions regarding our manuscript “Can circulating MicroRNAs, Cytokines, and Adipokines help differentiate Psoriatic Arthritis from Erosive Osteoarthritis of the Hand? A Case-Control Study.” These suggestions have significantly improved our paper. We addressed your concerns and hope that you will now find the revised article more suitable for publication in the IJMS. Below, you will find a point-by-point answer to your comments. All changes have been highlighted in red color in the revised manuscript.

Detailed responses to Reviewer #2

Point-by-point response to Comments and Suggestions for Authors

Q1: The study does not specify the number of patients, raising concerns about statistical data.

A1: We apologize for the oversight. The number of subjects is now explicitly stated in the abstract. It was previously reported in the Tables and main text.

Q2: Findings should be validated in an independent cohort to confirm biomarker reliability.

A2: The Discussion clearly emphasized the necessity of external validation in a prospective cohort. However, collecting such data will necessitate significant time, and we sincerely hope that other research groups, beyond our own, will be inspired by our findings to undertake this endeavor.

Q3 and Q6: Were comorbidities (obesity, diabetes) or medications accounted for, as they influence adipokines/cytokines? / Did the study rule out RA or other arthritis types that could confound miRNA/cytokine profiles?

A3 and A6: Following our previous work, we accounted for comorbidities using bivariable logistic regression, as reported in Supplementary Tables. We rewrote the sentences about the exclusion criteria, ensuring that they align with the information provided in our previous work (Cheleschi S et al. Circulating Mir-140 and leptin improve the accuracy of the differential diagnosis between psoriatic arthritis and rheumatoid arthritis: a case-control study. Transl Res. 2022;239:18-34. 10.1016/j.trsl.2021.08.001).

Q4: Peripheral blood mononuclear cells (PBMCs) include mixed cell populations; cell-specific miRNA changes may be diluted.

A4: We concur with the Reviewer’s observations: the methods used for collecting and isolating PBMCs can impact the integrity of nucleic acids, and it’s crucial to minimize the “dilution effect.” We rephrased the “Laboratory analysis” section and added relevant references.

Q5: ELISA measures total protein levels but may miss post-translational modifications affecting cytokine activity.

A5: We thank the reviewer for his/her interesting comment, but the aim of the present study was to evaluate the potential role of a panel of cytokines as biomarkers to differentiate EHOA to PsA.

Q7: The study shows associations but does not demonstrate if miRNAs/cytokines play a causal role in disease mechanisms.

A7: We concur with the reviewer’s perspective, but the primary objective of this study was not explanatory but predictive. The aim was to identify potential serum markers that could aid in differentiating EHOA from PsA. See for instance Steyerberg EW et al. Assessing the performance of prediction models: a framework for traditional and novel measures. Epidemiology. 2010;21:128-138. 10.1097/EDE.0b013e3181c30fb2 and Efron B. Prediction, Estimation, and Attribution. J Am Stat Assoc. 2020;115:636-655. 10.1080/01621459.2020.1762613.

Q8: The discriminatory power of miR-155 + CRP needs AUC/ROC analysis to assess clinical utility.

A8: Supplementary Table 1s provide AUCs for all univariable and multivariable predictors.

Q9: Were there PsA/EHOA patients with similar biomarker profiles, reducing diagnostic accuracy?

A9: If we interpret this question accurately, the answer is that our sample did not contain patients with identical biomarker profiles.

 Q10: Do these biomarkers change with disease progression or treatment?

A10: This is certainly an intriguing question. However, we cannot answer it using the current study design (case-control). We have, however, expanded the discussion section to include reports of previous experiences (Ciancio G et al. Clin Exp Rheumatol. 2017;35:113; Giannitti C. et al. Int J Biometeorol. 2017; 61:2153).

Q11: Higher TNF-α in EHOA contradicts its known role in PsA; is this due to differences in disease activity?

A11: We concur with the Reviewer that the higher levels of TNF-α in EHOA compared to PsA are surprising. We can speculate that this finding may be attributed to the relatively low inflammatory activity observed in our PsA patients. Unfortunately, we did not conduct any ultrasound or magnetic resonance imaging studies to evaluate the presence and severity of inflammation in both patient populations. This is now acknowledged as a study limitation.

Q12: Extracellular miRNAs can degrade; were pre-analytical conditions strictly controlled?

A12: We thank the Reviewer for this important observation. We have added the following sentences into Discussion section:

“In recent years, circulating miRNAs have emerged as potential biomarkers for diagnosing and predicting the prognosis of various pathological conditions, including cardiovascular, neurological, oncological, metabolic, and rheumatological diseases.  In fact, miRNAs are highly stable in biological fluids and resistant to endogenous ribonucleases activity, as well as to extreme conditions such as high temperatures, long-term storage, and repeated freeze-thaw cycles. Additionally, circulating miRNAs are easily accessible and can be measured with high sensitivity. [21,26,34,37-40].”

Q13: miRNA qPCR data require proper reference genes; were stable controls used?

A13: We appreciate the reviewer’s valuable suggestion. We have revised the last sentences of the section “MiRNAs and cytokines expression analysis” to include the correct references.

Q14: Were corrections (e.g., Bonferroni) applied given the number of miRNAs/cytokines tested?

A14: Between-group comparisons, as presented in Table 1, may apply correction for three groups (PsA, EHOA, and HS). However, these comparisons are not relevant for identifying predictors in univariate or bivariate logistic regression analyses with PsA vs. EHOA as outcome. We identify potential predictors based on discrimination and calibration, see for instance Steyerberg EW et al. Assessing the performance of prediction models: a framework for traditional and novel measures. Epidemiology. 2010;21:128.

Q15: Blood biomarkers may not reflect local joint pathology; were synovial samples analyzed?

A15:We agree with the reviewer’s observation, but the primary goal of the study was to identify potential serum markers that could aid in differentiating between EHOA and PsA. Furthermore, we must acknowledge the challenge of obtaining sufficient quantities of synovial fluid from the small joints of the hands.

Q16: Adipokines (e.g., leptin) fluctuate with metabolic state; were fasting samples collected?

A16: Thank you for this hint. As reported in our previous study, blood samples were drawn in the supine position in the morning after an overnight fast  (Cheleschi S et al. Transl Res. 2022;239:18). We have added this information in the corresponding Section of Material and Methods.

Q17: miRNA profiling + ELISA may be expensive for routine diagnostics compared to imaging/clinical criteria.

A17: The need for a formal cost-benefit analysis is now discussed in the Discussion section.

Q18: Lab-to-lab variability in miRNA quantification could limit widespread adoption.

A17 and A18: We appreciate the reviewer’s insightful comments. To address their concerns, we’ve added two new sentences in the discussion section to highlight some practical limitations of miRNAs as biomarkers in clinical practice.

Q19: How do miR-155 and TNF-α interact in EHOA vs. PsA pathogenesis?

A19: This is a very intriguing question. As we discussed in the Discussion, miR-155 plays a crucial role in driving chronic inflammation and regulating immune cells, including T-cells, B-cells, and dendritic cells. Moreover, miR-155 stimulates dendritic cells to produce specific cytokines, such as TNF-α, IL-6, and IL-17, which are essential for the development of inflammatory T cells. While its role in the pathogenesis of psoriasis has been extensively studied, there is currently no data on its pathogenetic role in psoriatic arthritis.

Q20: Even if biomarkers differ, does this guide treatment decisions (e.g., anti-TNF for EHOA)?

A20: Your observation is accurate and pertinent, but the data from our study is preliminary and necessitates further investigation. In light of this, we have added new sentences in the Discussion Section along with a new reference (Tenti S et al. Ther Adv Musculoskelet Dis. 2023;15:1759720X231158618.)

Reviewer 3 Report

Comments and Suggestions for Authors

I would like to congratulate the authors for their interesting work on the differentiation between Psoriatic Arthritis and Erosive Hand Osteoarthritis. The study is well-structured, with a clear and logical layout. However, I would like to suggest the following revisions to improve the manuscript:

  • Line 6 – Please correct the typographical error "Independent Re and and rcher"
  • Please insert the number of patients and controls included in the study in the Abstract for clarity.
  • Additionally, it would be helpful to mention the results regarding IL-6 and IL-1β in the Abstract, even if they were not statistically significant, to provide a complete overview of the findings.
  • Lines 55–60 – At reference [16], where the CASPAR criteria are mentioned, I recommend that the authors briefly expand on the key features of the CASPAR classification criteria (e.g., inclusion of current psoriasis, personal or family history of psoriasis, dactylitis, nail dystrophy, negative rheumatoid factor, radiographic evidence of new bone formation). This would make the text clearer, especially for readers who may not be fully familiar with these criteria.
  • Line 254 – Where the authors state that the findings align with previous reports, please add the phrase "implying other localizations" to clarify the possible broader relevance of miR-155 dysregulation beyond hand involvement.

Author Response

General response to Reviewer #3

Many thanks for your constructive comments and valuable suggestions regarding our manuscript “Can circulating MicroRNAs, Cytokines, and Adipokines help differentiate Psoriatic Arthritis from Erosive Osteoarthritis of the Hand? A Case-Control Study.” Your insights have significantly improved our paper.

We have addressed your concerns and hope that you will now find the revised article more suitable for publication in IJMS. Below, you will find a point-by-point answer to your comments.

All changes have been highlighted in red color in the revised manuscript.

Detailed responses to Reviewer #3

I would like to congratulate the authors for their interesting work on the differentiation between Psoriatic Arthritis and Erosive Hand Osteoarthritis. The study is well-structured, with a clear and logical layout. However, I would like to suggest the following revisions to improve the manuscript:

Q1: Line 6 – Please correct the typographical error “Independent Re and and rcher”

A 1: Thank you for finding this typo. We’ve corrected the error.

Q2: Please insert the number of patients and controls included in the study in the Abstract for clarity.

A2: Thank you for pointing that out. We’ve added the number of patients to the abstract.

Q3: Additionally, it would be helpful to mention the results regarding IL-6 and IL-1β in the Abstract, even if they were not statistically significant, to provide a complete overview of the findings.

A3: We appreciated the reviewer’s comment and made the necessary modifications to the sentences in the Abstract Section.

Q4: Lines 55–60 – At reference [16], where the CASPAR criteria are mentioned, I recommend that the authors briefly expand on the key features of the CASPAR classification criteria (e.g., inclusion of current psoriasis, personal or family history of psoriasis, dactylitis, nail dystrophy, negative rheumatoid factor, radiographic evidence of new bone formation). This would make the text clearer, especially for readers who may not be fully familiar with these criteria.

We appreciate the reviewer’s thoughtful suggestion and have included the information about the CASPAR criteria in the revised version of the MS.

Q5: Line 254 – Where the authors state that the findings align with previous reports, please add the phrase "implying other localizations" to clarify the possible broader relevance of miR-155 dysregulation beyond hand involvement.

A5: In agreement with your suggestion, we have highlighted the relevance of miR-155 in the development and progression of osteoarthritis (OA) not only in the hand but also in other joints.

Round 2

Reviewer 2 Report

Comments and Suggestions for Authors

Authors have successfully answered all my comments